# Using Optimal Estimation to Retrieve Winds from VAD Scans by a Doppler Lidar

Sunil Baidar[1,2], Timothy J. Wagner[3], David D. Turner[4], W. Alan Brewer[2]

[1]Cooperative Institute for Research in Environmental Sciences, University of Colorado Boulder, Boulder, CO 80309, USA
5   [2]Chemical Sciences Laboratory, National Oceanic and Atmospheric Administration, Boulder, Colorado 80305 USA
[3]Space Science and Engineering Center, University of Wisconsin-Madison, Madison, Wisconsin 53706 USA
[4]Global Systems Laboratory, National Oceanic and Atmospheric Administration, Boulder, Colorado 80305 USA

*Correspondence to*: Sunil Baidar (sunil.baidar@noaa.gov)

10   **Abstract.** Low-powered commercially-available coherent Doppler lidar (CDL) provides continuous measurement of vertical profiles of wind in the lower troposphere, usually close to or up to the top of the planetary boundary layer. The vertical extent of these wind profiles is limited by the availability of scatterers, and thus varies substantially throughout the day and from one day to the next. This makes it challenging to develop continuous products that rely on CDL-observed wind profiles. In order to overcome this problem, we have developed a new method for wind profile retrievals from CDL that combines the traditional velocity-azimuth display (VAD) technique with optimal estimation (OE) to provide continuous wind profiles up to 3 km. The new method exploits the level-to-level covariance present in the wind profile to fill in the gaps where the signal to noise ratio of the CDL return is too low to provide reliable results using the traditional VAD method. Another advantage of the new method is that it provides the full error covariance matrix of the solution and profiles of information content, which more easily facilitates the assimilation of the observed wind profiles into numerical weather prediction models. This method was tested using a yearlong CDL measurements at the Atmospheric Radiation Measurement (ARM) Southern Great Plains (SGP) Central Facility in 2019. Comparison with the ARM operational CDL wind profile product and collocated radiosonde wind measurements shows excellent agreement ($R^2 > 0.99$) with no degradation in results where the traditional VAD provided a valid solution. In the region where traditional VAD does not provide results, the OE wind speed and wind vector have uncertainties of 3.44 and 4.33 m/s respectively. As a result, the new method provides additional information over the standard technique and increases the effective range of existing CDL systems without the need for additional hardware.

## 1 Introduction and Background

The kinematic profile of the planetary boundary layer (PBL) has a significant impact on disciplines throughout the atmospheric sciences. Low-level wind shear can determine storm mode (e.g. Davies and Johns, 1993) and has significant impacts on aviation safety (e.g. Gultepe et al., 2019; Thobois et al., 2018), while knowledge of the wind profile within the PBL is a significant factor in siting wind energy installations (Banta et al., 2013). High-temporal resolution observations of the wind

profile are crucial for understanding numerous atmospheric processes. While radiosondes remain the standard by which all profiling measuring systems are evaluated, they are not well-suited toward capturing the evolution of boundary layer wind profiles due to their substantial cost per observation and significant time required to prepare and execute each observation. Alternative ways of observing atmospheric wind profiles have been developed, including active remote sensing with radars, lidars, and sodars; passive remote sensing with satellites; and in situ observations with commercial aircraft or uncrewed aircraft systems (UAS).

To address the need for rapid sampling of the wind profile in the PBL, manufacturers have developed low-powered commercial coherent Doppler lidar (CDL) wind profilers. These systems feature turnkey operation, are quick and easy to deploy, and have the ability to run unattended for significant periods of time. Fundamentally, Doppler lidars measure the velocity of scatterers along the emitted beam (radial velocity or line of sight velocity, LOSV); it is assumed that the one-dimensional speed of the scatterers is the same as the wind speed along that direction since the primary scatterers are aerosols. Observations of the vertical profile of the horizontal or three-dimensional wind vector can be retrieved from CDL-observed radial velocities using techniques like velocity-azimuth display (VAD) or Doppler beam swinging (DBS). In both of these techniques, lidar measurements along multiple non-coplanar angles are used to reconstruct the vertical profile of the wind vector under the assumption that winds are horizontally homogeneous within the volume observed by the lidar and that they do not evolve during the period (usually a minute or less) in which a set of scans is collected. Numerous studies comparing Doppler lidar wind profile retrievals to in situ observations from radiosondes or instrumented towers and masts have shown that CDLs are a reliable and effective way of measuring wind profiles in the PBL (e.g. Choukulkar et al., 2017; Klein et al., 2015).

While the theoretical maximum range of CDLs is 10 km or more and is only limited by the pulse repetition frequency of the emitter and the number of range gates in the detection system, the need for the signal to be scattered and returned to the lidar means that the effective range is much less. CDLs usually feature a laser emitting at 1.5 μm. This wavelength is short enough to be sensitive to aerosols, cloud droplets, and some precipitation, but not so short that it is significantly impacted by molecular scattering. This means it can be a challenge to obtain wind observations at times and heights where aerosol content is low, such as above the top of the PBL. In practice, CDL-observed wind profiles usually extend to 1-2 km above ground level (AGL). While this observation depth is more than sufficient for wind energy applications, other processes such as PBL entrainment or mesoscale dynamics extend to higher altitudes and are difficult to assess with operational CDL retrievals. Furthermore, since the aerosol concentration is not constant, the maximum effective height of CDL-observed wind profiles varies substantially throughout the day and from one day to the next. This makes it challenging to develop continuous products that rely on CDL profiles as the valid range is constantly changing. Various techniques have been developed to extend the range of wind profiles from scanning CDL, including accumulation of signal power spectra estimates for direct estimation of the wind vector without estimating radial wind velocities for individual azimuth angles (Smalikho, 2003; Stephan et al., 2019). Although these advanced techniques are able to extract information from noisier Doppler spectra, they are still limited by the availability of the scatterers and hence, do not provide consistent vertical coverage.

In the present work, we propose an alternate method of retrieving wind profiles from CDL observations that combines the traditional VAD technique with optimal estimation (Rodgers, 2000). This exploits the level-to-level covariance present in the wind profile to help fill in the gaps where the signal to noise ratio (SNR) of the lidar return is not strong enough to perform the traditional VAD technique. The output of this retrieval technique is a near-continuous profile of winds up to 3 km AGL that agrees very strongly with the traditional VAD at times and heights where both are available, yet still exhibits strong agreement with radiosondes at heights where the traditional VAD technique was unable to produce a valid result. The remainder of this paper discusses the retrieval methodology (Sect. 2), compares its performance against both the traditional VAD and collocated radiosondes (Sect. 3), discussion (Sect. 4) and offers recommendations and conclusions (Sect. 5).

## 2 Methodology

### 2.1 Traditional VAD Method (VADtrad)

In the traditional VAD method (VADtrad), horizontal winds are retrieved from scanning CDL plan position indicator (PPI) or step-stare scans at one or multiple elevation angles (EAs) using the VAD algorithm described by Browning and Wexler (1968). The measured radial velocity $y_r$, at a given range gate $r$ is related to the three-dimensional wind velocity vector $x_r$ by the viewing geometry. Assuming a horizontally homogeneous wind flow and constant vertical velocity over the sampling volume, a sinusoid is fitted to the radial velocity data at a given range gate (or range bin) to retrieve the wind velocity components. The wind speed, wind direction and vertical velocity are provided by the amplitude, phase, and offset of the sinusoid, respectively. Details of VADtrad retrievals and wind precision estimates from a CDL can be found in Newsom et al. (2017). Briefly, for $N$ number of beams with azimuth angles ($\theta_i$) in the PPI or step-stare scans at elevation angle ($\alpha$), and measurement uncertainty due to random errors ($\sigma$), the VADtrad method is equivalent to minimizing

$$\psi^2 = \sum_{i=1}^{N} \frac{\left(x_r f_i^T - y_{ri}\right)^2}{\sigma_{ri}^2} \tag{1}$$

With $f_i = [sin\theta_i \cdot cos\alpha \quad cos\theta_i \cdot cos\alpha \quad sin\alpha]$ representing the measurement geometry of individual beams.

### 2.2 Optimal Estimation VAD Method (VADoe)

Wind velocity components are retrieved one range gate at a time and hence one height at a time from a set of radial velocity measurements from an azimuthal scan at a given range gate with the VADtrad technique. While this level-by-level retrieval can filter out individual bad radial velocity data at each level by applying SNR thresholds or multiple passes of the sinusoidal fit to determine outliers, it ignores the level-to-level correlation in wind velocity that exists in the atmosphere, information that can be used to inform about the characteristics of the wind profile further away from the surface. Figure 1 shows the correlation matrices for the $u$ and $v$ component of wind vectors calculated from radiosonde measurements at the Atmospheric Radiation Measurement (ARM) Southern Great Plains (SGP, Sisterson et al., 2016) Central Facility (C1) in north-central Oklahoma for the month of July. These correlations were calculated from covariance matrices compiled from 15 years of radiosonde data

(2004-2019) from the ARM facility (SONDEWNPN, 2001), which usually launches radiosondes every six hours. Since the correlation matrices are symmetric about the diagonal, the lower-right half of the panels in Fig. 1 have been replaced with the correlation matrix for a single representative retrieval for a clear-sky day in July 2019 (to be discussed later). It is clear that very strong correlations in the prior dataset (i.e., above the diagonal in Fig. 1) exist for wind components at adjacent heights, while heights that are separated by hundreds of meters still exhibit correlations of 0.5 or more. This information can be used to assist in retrieving the wind profile at higher altitudes where the lidar SNR is low, provided that a sufficient number of observations are available from other sources, such as radiosondes or aircraft, to generate the covariance matrices.

One way of integrating the level-to-level correlations with CDL radial velocity observations to produce continuous wind profiles is through the implementation of an optimal estimation retrieval (OE, Rodgers, 2000). In optimal estimation, a set of measurements $y$ is related to the state vector $x$, which contains parameters describing the current atmospheric state, by a forward model $F$:

$$y = F(x, b) + \epsilon \tag{2}$$

where $b$ represents model parameters that are not retrieved and $\epsilon$ represents the model error. In essence, the forward model maps the state of the atmosphere to a set of variables that can be observed directly and contains the physical and instrumental factors that describe the measurements. For many remote sensing applications, the forward model is a radiative transfer model that converts the state of the atmosphere (such as profiles of temperature, water vapor, and trace gases) to radiances at various wavelengths measured by satellites or ground-based radiometers. Through the optimal estimation technique, this relationship is inverted so that a set of observations can be used to obtain the atmospheric state. The optimal estimation technique has been extensively used for retrievals of atmospheric constituent profiles from passive remote sensing measurements where the problem is generally ill-determined (e.g. Kuang et al., 2002; Maahn et al., 2020; Turner and Blumberg, 2019; Turner and Löhnert, 2014). Since ill-determined problems can produce an infinite number of solutions, *a priori* information in the form of the mean and covariance of the state vector is used as a constraint to help the algorithm obtain a solution that is both physically possible and statistically likely to occur for a particular location and time of year.

In the present case, in which scanning CDL measurements of radial velocities at different azimuth ($\theta$) and elevation ($\alpha$) angles are being used to obtain the components of the wind vector ($u$, $v$, and $w$), the forward model is simply the geometry of the measurement that maps the wind vector to the radial coordinate system. It is given by:

$$F = [sin\theta \cdot cos\alpha \quad cos\theta \cdot cos\alpha \quad sin\alpha] \tag{3}$$

If one assumes that the vertical velocity, $w$ is much smaller than the horizontal velocity then the contributions of $w$ to the radial wind vector can be neglected. The forward model then reduces to:

$$F = [sin\theta \cdot cos\alpha \quad cos\theta \cdot cos\alpha \;] \tag{4}$$

Since the forward model $F$ is independent of the state vector $x$, the jacobian $K$ of the forward model $F$ with respect to the elements of the state vector $x = [u, v]$ is also the forward model

$$K = \frac{dF}{dx} = F \tag{5}$$

Equation (2) can then be linearized as

$$y = Kx + \epsilon \tag{6}$$

The maximum a posteriori solution for Eq. (6) is

$$x = x_a + (K^T S_\epsilon^{-1} K + S_a^{-1})^{-1} K^T S_\epsilon^{-1}(y - Kx_a) \tag{7}$$

where $x_a$ is the *a priori* profile and $S_a$ and $S_\epsilon$ are the *a priori* and measurement error covariance matrices, respectively. The VADoe retrievals are performed on a fixed vertical resolution defined by the range gate size of the DL measurement. Note that Eq. (6) has an analytical solution, which is the VADtrad result, but provides an unreasonable solution when measurement SNR is low. This is the reason the VADtrad algorithm is performed layer-by-layer and a SNR threshold is applied. Since the

present work evaluates the VADoe retrieval at the ARM SGP Central Facility, the *a priori* information is calculated from 15 years of profiles of wind speed and direction observed by radiosondes launched at that site to create monthly mean $u$ and $v$ profiles ($x_a$) and covariances ($S_a$). By using monthly *a priori* information instead of a single priori dataset that spans all seasons for all retrievals, natural variation in the winds can be captured. Few locations will have the in situ observational density that the ARM SGP site does, but alternate sources of *a priori* data could include Airborne Meteorological Data Relay (Moninger

et al., 2003) observations or model output.

Radial velocity uncertainty ($\sigma_r$) for a range gate is estimated by calculating the mean of the variance of radial velocity over two neighboring range gates for each azimuthal stare. For a VAD scan with $n$ azimuth angles, the radial velocity uncertainty for the j$^{th}$ range gate is given by

$$\sigma_r^2(r_j) = \frac{1}{3n} \sum_{i=1}^{i=n} \sum_{k=j-1}^{j+1} \left( y_r(\theta_i, r_k) - \overline{y}_r(\theta_i, r_j) \right)^2 \tag{8}$$

where

$$\overline{y}_r(\theta_i, r_j) = \frac{1}{3} \sum_{k=j-1}^{k=j+1} y_r(\theta_i, r_k) \tag{9}$$

This formulation is similar to the Trial 2 radial velocity uncertainty formulation given in Newsom et al. (2017), where radial velocity uncertainty is calculated over consecutive scans and neighboring range gates. This formulation was found to result in the best agreement between wind speed and direction precision estimates from the VADtrad algorithm and sonic anemometer

measurements from the collocated 300 m tower at the Boulder Atmospheric Observatory during the eXperimental Planetary boundary-layer Instrument Assessment (XPIA) field campaign (Lundquist et al., 2017). Unlike Newsom et al. (2017), the formulation given by Eq. (8) assumes isotropy in atmospheric variance for a given range gate. Because Eq. (8) assumes isotropy and ignores SNR dependency of measurement uncertainty, the instrument noise component $\sigma_n$ is added to $\sigma_r$ to compute total measurement error. Figure 2 shows the CDL radial velocity precision as a function of SNR determined from ARM SGP C1

Doppler lidar vertical stare measurements using the method described in Lenschow et al. (2000) and available as part of the standard ARM vertical velocity statistics dataset (Newsom, R. K., Sivaraman, C., Shippert, T. R., Riihimaki, 2019a).

The total measurement error variance $\sigma_\epsilon$ for a given viewing geometry $i$ used for constructing the measurement error covariance matrix is then given by

$$\sigma_{\epsilon_i}^2 = \sigma_r^2 + \sigma_{n_i}^2 \tag{10}$$

Both the maximum possible $\sigma_r$ and $\sigma_n$ are limited by the CDL measurement bandwidth (±19 m/s for the SGP lidar) and is much smaller in magnitude than the variability described by the *a priori* covariance (see Fig. 1). This results in the measurement being artificially weighted higher even when the measurement has little to no information (region to the left of the dashed vertical line in Fig. 2). In order to overcome this, $\sigma_n$ are set to a large number (100 m/s) for SNR below 0.005 (~-23 dB). This value needs to be optimized depending upon the number of azimuth beams used in the retrieval. The non-diagonal elements

were set to 0, assuming there was no correlation between the uncertainties at different range gates and different azimuth angles. The solution given by Eq. (7) is a weighted mean of the *a priori* profile and the information from the measurement. The weight is given by the averaging kernel matrix $A$,

$$A = (K^T S_\epsilon^{-1} K + S_a^{-1})^{-1} K^T S_\epsilon^{-1} K \tag{11}$$

The retrieval at any range gate is an average of the whole profile weighted by the row of the averaging kernel matrix

corresponding to that range gate. The $A$ matrix also can be used to determine the number of independent pieces of information retrieved (often quantified as the degrees of freedom, or DOF) as well as an estimate of the vertical resolution of the retrieved profile at a given level. Note that in a traditional VAD level-by-level retrieval, each range gate is considered as independent and the range gate resolution defines the vertical resolution of retrieved profiles. For an ideal retrieval scenario, $A$ is the identity matrix, the DOF equals the number of retrieved profile layers and the averaging kernels peak at their corresponding altitudes.

In reality, the retrieved profile is a smoothed version of the true profile. In case of the scanning CDL measurements, $A$ is close to an identity matrix throughout most of the PBL where the SNR is relatively high. As a result, the *a priori* provides minimal to no constraint at those altitudes. However, at altitudes where measurement SNR is low, the VADoe retrieval is capable of providing an *a priori*-constrained retrieval, even below the SNR threshold usually applied to VAD retrievals. In cases when there are very few or no valid CDL measurements (e.g. a very low aerosol loading, foggy or rainy day), the retrieved profiles

are only constrained by the a priori and hence, follows the a priori profile. Such profiles can easily by identified using $A$ matrix, DOF or retrieval error.

    One of the advantages of the optimal estimation technique is that uncertainties from both the instrument and the retrieval are propagated throughout the process so an overall error for each individual observation can be easily quantified. The posterior error covariance matrix which includes contributions from smoothing error and measurement error is given by

$$S_{op} = (S_a^{-1} + K^T S_\epsilon^{-1} K)^{-1} \tag{12}$$

    An additional source of uncertainty is the accuracy of the forward model, which is affected by the assumption of horizontally homogeneous wind flow, isotropic turbulence and vertical velocity component is negligible. This forward model error is given by

$$S_f = G_y \Delta f^2 G_y^T = G_y [f(x, b, b') - F(x, b)]^2 G_y^T \tag{13}$$

where $G_y = AK$ is the gain matrix, $f$ is the idealized forward model which includes all the correct physics and $F$ is the simplified approximation. In an ideal scenario, $f(x, b, b') = y$ . Therefore, we calculated forward model error as

$$S_f = G_y \Delta f^2 G_y^T = G_y [y - Kx]^2 G_y^T \tag{14}$$

The total retrieval error covariance $S_{total}$ is given by

$$S_{total} = S_{op} + S_f \tag{15}$$

The square root of the diagonal element of $S_{total}$ provides the 1-σ uncertainty for the retrieved $u$ and $v$ profiles.

## 3 Results

ARM operates a total of 5 CDLs at the SGP site: one Halo Streamline XR at the C1 facility and four Halo Streamlines (Pearson et al., 2009) at extended facilities that surround the C1 site at a distance of approximately 50 km. Each ARM CDL makes near-continuous measurements of radial velocity and attenuated backscatter coefficient profiles at a wavelength of 1.5 μm. These

CDLs are sensitive to aerosols but not molecular backscatter and hence the measurements are confined to the PBL. Details about the ARM Doppler lidars, their operations, and data products are found in Newsom and Krishnamurthy (2020). We used the CDL measurements at the SGP C1 (DLPPI, 2010) for this study. The Doppler lidar at the SGP C1 site that is further examined in this work operates with 30 m range gate resolution and 1.3 s time resolution. It is typically configured to perform one 8-beam PPI scan at a 60 degree elevation angle every 15 minutes and performs vertical stares during the remaining time.

The PPI scan takes approximately 40 seconds. Horizontal wind profiles are retrieved from the PPI scan using the VAD method. The ARM DL wind retrieval algorithm is described in detail in Newsom et al. (2019b). It employs a SNR threshold of 0.008 (~ -21 dB) to filter out poor-quality radial velocity data before computing wind profiles.

To evaluate the performance of the optimal estimation retrieval against real-world observations, VADoe retrievals from the ARM SGP C1 lidar were processed for the entirety of the 2019 calendar year. Normally, radiosondes are launched

from the SGP site four times a day, but the launch frequency was doubled to eight daily sondes for the three-month period lasting from May through July 2019. In all, over 1600 radiosondes were collocated with Doppler lidar profiles during the year-long study period. Each radiosonde was temporally matched to the Doppler lidar profile that was taken nearest in time to the radiosonde launch time. Radiosondes that were launched more than 30 min from the nearest valid lidar observation were excluded from this analysis.

An important parameter for evaluating the utility of an optimal estimation retrieval is the information content. One measure of this is the degrees of freedom of the signal (DFS) which can be used to identify how many unique pieces of information are present in the retrieval as well as determine at what altitudes the information can be found. While the OE wind retrieval is output onto a fixed grid with 113 evenly-spaced levels from the surface to 3000 m, the fact that OE-retrieved observations are overlapping weighted averages of various depths in the atmosphere means that there will be fewer than 113

uncorrelated pieces of information in the output. The cumulative DFS of the retrieval at the nth level is the sum of the first n elements of the diagonal of the averaging kernel $A$ (Eq. 10); the total DFS of the retrieval is thus simply the trace of A. Figure 3 shows the mean and 25[th] and 75[th] percentile of the cumulative DFS as a function of height for the 1600+ OE wind profiles that were matched to a radiosonde. On average, the total profile DFS is approximately 15 though the variability ranges from 9.5 to 18.5, and the $u$ and $v$ DFS are effectively identical. Most of the DFS are concentrated in the lowest 1000 m, with

approximately 10.3 DFS on average below that height which means the true vertical resolution of the DL is around 100 m. True vertical resolution of the DL wind profiles can be improved by including multiple PPI at different EAs and increasing the number of azimuth angles in a PPI scan. However, with roughly 5 DFS in the OE retrieval above 1000 m, the retrieval can still provide valuable information about an otherwise under observed layer of the atmosphere. An advantage of calculating the cumulative DFS and the related true vertical resolution profiles from the optimal estimation retrieval is that it easily facilitates the assimilation of the observed wind profiles into NWP (Coniglio et al., 2019).

One way to evaluate the performance of the OE winds is by examining a sample plot of the winds as measured by various systems. Figure 4 depicts time-height cross sections of the *v* component of the wind on 16 May 2019. This was a quiescent day at the SGP site with a persistent upper-level ridge ensuring few clouds and little synoptic forcing. These conditions enabled the formation of a low-level jet (LLJ) over the region, with winds approaching 20 m s$^{-1}$ approximately 250 m above the ground at 0600 UTC (1:00 AM local time). Since this was during the period of 3 h radiosonde launches from SGP, enough radiosonde profiles are present to capture some of the short-term variability in the atmospheric state. The VADtrad profiles are limited to heights approximately 1000 m AGL and below. While VADtrad can resolve the LLJ and daytime turbulence in the PBL, an insufficient number of scatterers above those levels means that the VADtrad is incapable of resolving any phenomena at higher altitudes. By contrast, the OE provides continuous profiles from the surface to 3000 m AGL. While the information content is not as large at these higher altitudes as noted previously, the presence of even a few independent data points in the 1500 - 3000 m range can bring new insight to processes in the entrainment zone and free troposphere. For example, the sondes indicate a secondary maximum of *v* winds above the low level jet between 1000 and 2000 m. The 0.008 SNR threshold used operationally by ARM means that this feature is missed entirely by VADtrad. Likewise, in the afternoon hours (after 1900 UTC) the PBL has grown too deep to be fully resolved by the VAD, yet the OE retrieval is able to monitor the continued increase in the depth of the turbulent winds as it allows even regions of low SNR to be used and to have an impact on the retrieved profile. The sondes are able to note the depth of this layer, but the 3 h launch frequency is still too coarse to resolve the individual elements the way the OE retrieval can. Note that radiosonde profiles shown in Fig. 4c are interpolated in time for illustration purposes. OE results in Fig. 4b shows faint vertical striping at higher altitudes where there is little to no information available from DL. This is due to the inherent nature of the VADoe retrieval which includes level-to-level correlation but no time dependent information. Results at higher altitudes are more influenced by nearest good measurements compared to further away. The profile-to-profile difference at higher altitudes are within VADoe retrieval error for most cases as shown in Fig. 4d.

### 3.1 Radiosonde comparisons

To facilitate intercomparisons between the radiosondes and both VADtrad and VADoe, the same vertical grid from the traditional VAD technique was used for the OE output, and the radiosonde observations were averaged to that grid. Quality control measures included rejecting VAD observations with an absolute value greater than 50 m s$^{-1}$ and OE retrievals where

the OE-derived measurement uncertainty exceeded 5 m s$^{-1}$. Note that due to the stringent SNR threshold (<-21 dB) applied to the VADtrad data from the ARM database, there were no VADtrad observations with uncertainty greater than 5 m s$^{-1}$.

Scatter plots showing the performance of both the traditional VAD-derived CDL wind observations and the OE-retrieved CDL winds throughout the 2019 analysis period are shown in Fig. 5. Several important points emerge from this figure. First, it is important to note that the VAD and OE wind observations are almost identical for the times and heights where both are available as the correlation coefficients between the two sets of CDL observations are 0.998 and 0.999 for the $u$ and $v$ wind components respectively. In essence, using the VADoe retrieval in place of the VADtrad technique does not degrade the

quality of the observations but instead augments existing observations with additional information at heights above those observed by VADtrad. Second, the VADtrad winds appear to have a stronger correlation with the radiosondes than the VADoe winds do at first glance. However, the OE winds include many observation points where the traditional technique does not provide an observation (N = 139,582 for OE and 59,403 for VAD). A more appropriate analysis limits the intercomparison to only the points that are present in both VADoe and VADtrad. In those cases (depicted with orange points in Fig. 5b and 5e),

the sonde/OE correlations are functionally identical to the sonde/traditional comparisons and have effectively the same correlation values, and standard deviations (scatter). This further reinforces the idea that the OE winds can be used in place of the traditional VAD winds without degrading the near-surface observations. Finally, it is worth noting that, regardless of the instruments being compared, correlations are higher for the $v$ component than they are for the $u$ component. This may be due to the fact that the flow over the SGP site is persistently southerly and the $u$ wind tends to be more variable than the $v$ wind.

Note that natural variability in winds and turbulence results in an inherent scatter between lidar and sonde wind measurements.

### 3.2 Differences as a function of height

The mean and standard deviation of the lidar-minus-sonde differences at a given observation height can be used to determine the bias and uncertainty present in the lidar observations at that height. Figure 6 shows the vertical profile of the bias (mean

difference) and uncertainty (standard deviation of the differences) for both the VADtrad and VADoe profiles relative to the radiosondes throughout the lower troposphere. It is important to note that the differences between the VADtrad and VADoe methods seen here almost entirely due to a more comprehensive set of points being observed by the VADoe method. When only points that are valid for both methods are used (not shown), there is effectively no difference in the bias or standard deviation for either method at any height. This is expected given the extremely high degree of correlation between the

observed wind vectors as seen in Figs. 5c and 5f. Across the various panels in Fig. 6, it is clear that the VADoe retrievals are comparable to or better than the VADtrad winds at all analyzed heights, especially above typical PBL heights. With respect to the $u$ component (Fig. 6a), the two techniques have nearly indistinguishable performance in the lowest 500 m, as both have a slight slow bias that increases from -0.18 at the lowest range gate to -0.39 m s$^{-1}$ at 800 m. The uncertainties slightly increase over that depth, from 1.20 m s$^{-1}$ at the lowest level to 1.52 at 800 m. From 500 m to 1400 m, the bias remains similar for both

techniques, but the uncertainty starts to diverge as the VADoe $u$ uncertainty shows slightly larger values than the VADtrad $u$ uncertainty. Above 1400 m, substantial differences in the performance of the two systems are present. The VADoe $u$ wind bias

is negative but small, and increasingly approaching zero with increasing height, but the VADtrad $u$ bias becomes much more negative as height increases. The VADoe $u$ wind bias at 1500 m, 2000 m, and 3000 m is -0.58, -0.48, and -0.24 m s$^{-1}$ respectively; while the VADtrad $u$ bias at those heights is -0.95, -3.46, and -6.24 m s$^{-1}$ respectively. There is similar inflation in the uncertainty with height above 1300 m, as the VADoe technique continues its near-linear trend of increasing uncertainty with height while the VADtrad uncertainty shows marked increases once heights exceed 1600 m. While the $u$ bias was slow at all depths for both techniques, the $v$ bias is generally fast (Fig. 6b). Like the $u$ bias, the VADoe $v$ bias is small at all heights. It never exceeds 0.4 m s$^{-1}$ at any height while the VADtrad $v$ bias steadily increases throughout the analyzed depth to more than 3.6 m s$^{-1}$ at the highest levels. At the lowest levels (below 500 m) the VADoe $v$ uncertainties are again approximately the same as the VADtrad $v$ uncertainties. In the middle levels (between 500 and 1600 m) the VADoe uncertainties are larger, but there are many more valid points being included in the analysis as seen in Panel 6f. As noted above, the two methods agree almost perfectly when both are available, so the increase in uncertainty comes from points that the VADtrad is unable to observe at all. There is a local maximum in uncertainty below 500 m for the $v$ differences that is not present in the $u$ observations; this is likely an impact of the largely meridional low level jet frequently found over the SGP site after sunset.

The $u$ and $v$ components can also be used to calculate the vector difference, which is a convenient way of combining speed and direction error into a single parameter. These results are seen in Fig. 6c. The differences in bias in the lowest 1600 m are largely due to differences in observed wind speed, discussed further below. Above that height, the VADtrad again has larger biases and uncertainties than the VADoe product, largely due to the very small number of points included for analysis above those heights.

While the retrieval is conducted in terms of $u$ and $v$, it is instructive to evaluate how the retrieval performs in terms of wind speed and direction. These are presented in Fig. 6d and 6e respectively. The biases for both speed and direction are effectively identical for the two observation sets below 500 m with a value of approximately 0.5 m s$^{-1}$. Above that height, the VADoe observations show slightly more negative speed biases than the VADtrad observations do. Again, the speed uncertainty is slightly larger for the VADoe data until 1600 m at which point the VADtrad uncertainty increases rapidly. Direction differences show identical biases of approximately -5 deg until 1200 m, at which point the VADoe bias becomes less negative and starts to become positive while the VADtrad bias becomes markedly more negative with height.

Figure 6f also shows the number of valid intercomparisons as a function of height by showing the number of valid lidar/sonde intercomparisons for each lidar range gate. Here, it is clear how rapidly the number of VADtrad observations decreases with height due to the decreasing concentration of scatterers. At 1 km AGL, the number of observations is only 59% of what it was at the lowest range gate. By comparison, the VADoe retrieval still has 98% of the lowest-level observations. The number of VADoe observations decreases with height due to the imposition of the 5 m s$^{-1}$ gross error check noted earlier, but it is clear that the decrease in the availability of the VADoe product with increasing height is much less than it is for the VADtrad wind profiles.

**3.3 Differences as a function of SNR values**

Since a significant advantage of the OE retrieval is providing observations at altitudes for which no VADtrad data are available at standard values for SNR, it is worth looking specifically at the performance of the observations as a function of SNR. As noted in Fig. 5 and 6, most of the spread in the OE-minus-sonde differences is occurring for the levels where VADtrad observations are not available. Figure 7-9 illustrates differences between VADoe and radiosondes for wind speed (Fig. 7), wind direction (Fig. 8), and wind vector (Fig. 9) for four different bands of SNR. As noted above, the ARM standard cutoff for SNR is 0.008 which corresponds to approximately -21 dB, and VADoe implements a soft -23 dB cutoff by setting individual radial velocity error to 100 m/s for data with SNR below -23 dB. The data are divided approximately evenly into two groups with a higher SNR than the -21 dB cutoff (SNR1: > -13 dB, and SNR2: -21 to -13 dB), as well as SNR between -21 and -23 dB (SNR3), and SNR below -23 dB (SNR4). In order to minimize the impact of cloud returns, the highest SNR group (SNR1) is limited to data from lowest 800 m. Note that CDL SNR is not range corrected and hence absolute SNR threshold cannot be applied to filter for clouds. The performances of the first 3 SNR bands, including the SNR band in between the VADtrad and VADoe cutoff (SNR3, Panel c) are very similar. Wind speed biases (uncertainties) are -0.21 ms-1 (1.96), 0.05 ms-1 (1.56), and -0.11 ms-1 (1.51) respectively for SNR1, SNR2 and SNR3 groups. Similarly, wind direction biases (uncertainties) are -2.44 (47.4), -2.93 (46.4), and -1.50 (46.7) for the three highest SNR groupings. This comparable performance for those SNR groups are also highlighted in wind vector, which combines both the wind speed and direction differences (Fig 9). Slightly larger uncertainty for the highest SNR group (SNR1) is likely due to presence of higher variability in wind and higher turbulence in the lower PBL where the CDL SNR is greatest. Observation of precipitation droplets is another possible reasoning for the higher uncertainty and the tail in the distribution at the highest SNR bin. The comparable performance of the SNR3 group, which has SNRs in between VADtrad and VADoe cutoffs,  to the groups with better SNR indicates that at least some of the observations in this SNR region might potentially be available from VADtrad method if the SNR cutoff threshold were to be lowered. Nonetheless this also highlights the benefit of the VADoe retrieval where a more liberal SNR cutoff threshold could be applied.

Panel 7-9d shows observations that would not be available for VADtrad retrieval at all. As expected, both the bias and uncertainty are higher for this SNR group. The wind speed (direction) bias and uncertainty of this group of observations are -1.46 m/s (9.01˚) and 3.43 m/s (87.2˚) respectively. The wind vector RMSD for this SNR group is 4.3 m/s. While the VADoe observations in this SNR bands depict larger biases and greater uncertainty than the observations in better SNR bands, the wind speed uncertainty is comparable to the Tropospheric Airborne Meteorological Data Reporting (TAMDAR) system (Wagner and Petersen, 2021). The wind vector RMSD of less than 5 m/s, which is the error threshold used in the analyses, for this SNR group further supports that the VADoe retrieval errors are representative, and can be used to select data to meet different application requirements.   For example, VADoe data filtered for greater than 5 m/s error would meet the WMO threshold requirement for horizontal wind measurements in the free troposphere for Global and high resolution NWP (WMO, 2022). Considering that only 37.1% of the total dataset evaluated here has an SNR better than -21 dB, the VADoe technique provides many more usable observations.

## 4 Discussion

The comparisons in the previous sections show that VADoe provides identical results as VADtrad where VADtrad results are valid. At these levels, where most if not all of the information are coming from measurements, VADoe is mathematically equivalent to VADtrad. At lower SNR levels (or higher altitudes), where VADtrad results are not available, VADoe results compare favourably with radiosonde measurements. VADoe at those levels are statistically most likely output based on based on the (noisy) observations at those levels, higher quality (precision) measurements at lower levels and climatology. The VADoe retrieval provides well characterized uncertainty for each profile, and the corresponding averaging kernels allow the determination of both the vertical resolutions as a function of height and the maximum height to which the retrieval is mostly independent of the a priori profile. Thus, the retrieval errors and averaging kernels could be used to determine data that are suitable for a given application.

One of the biggest challenges of setting up the VADoe retrieval is appropriately scaling the CDL radial velocity measurement error at low SNRs to provide stable retrievals. The CDL radial velocity measurement error is limited by the measurement bandwidth. For example the measurement bandwidth for the ARM SGP CDL used here is +/- 19.4 m/s. This maximum measurement error is smaller than the a priori error (standard deviation). This becomes even smaller when you consider multiple radial velocities from different azimuths that are included in the retrieval. If the measurement errors are not inflated appropriately, measurements will always be weighted heavily compared to a priori, and results in unstable retrievals. Thus, measurement error at low SNR levels needs to be appropriately scaled accounting for number of azimuths and elevation angles included in the retrieval and magnitude of the a priori error.

Successful implementation of VADoe retrieval requires knowledge of the a priori mean profile and covariance. We used radiosonde measurements to create monthly mean profile and covariance. However, radiosonde measurement sites are limited which limits the applicability of the VADoe retrieval presented here to locations close of radiosonde sites. Future work should include testing using a priori from other sources such as AMDAR and NWPs. This would make VADoe retrieval more widely applicable, and also use of higher time resolution a priori in the retrieval.

## 5 Summary/Conclusion

Coherent Doppler lidars have many research, operational, and commercial applications. Through deployments around the world, they have proven to be reliable and robust instruments that have significantly enhanced our understanding of numerous processes and phenomena. However, since commercially available low-powered CDLs operating at 1.5 μm wavelength are insensitive to molecular scattering and thus must rely on aerosol scattering, the vertical extent of the wind profiles they observe is limited to the planetary boundary layer where aerosol concentrations are greatest. However, many key atmospheric processes are found at or above the top of the boundary layer which means that many CDLs are unable to observe them with standard algorithms consistently.

To provide profiles that are more vertically and temporally continuous, an optimal estimation retrieval was created so that established level-to-level correlations can be exploited to gain information about the wind profile at levels higher than those where CDLs can typically reach. This retrieval, called VADoe, is computationally simple as the forward model is derived from simple geometry. This method was tested using a yearlong CDL measurements in ARM SGP Central Facility in 2019. Comparison with collocated radiosonde and ARM operation CDL output (VADtrad) showed excellent agreement. Critically, with correlations of 0.998 and 0.999 between the VADtrad and VADoe for the $u$ and $v$ wind components respectively when both techniques are valid (i.e. the SNR in the observations is sufficient), using an OE retrieval does not degrade the existing retrievals. It merely provides additional information where none is currently available. The VADoe provides useful results, although with higher uncertainty, even when the SNR is too small and radial velocities are not reliable.

Optimal estimation retrievals have significant advantages for data assimilation. With well-characterized uncertainties for each observation and profiles of degrees of freedom of the signal and vertical resolution easily obtained as part of the retrieval, profiles from the VADoe algorithm are ready for assimilation into numerical weather prediction without needing to assume error profiles or other needed characteristics. Further, OE provides a framework for a combined wind profile retrieval from co-located different types of instruments for wind measurements (e.g. CDL, direct detection Doppler lidar, radar).

It is important to note that VADoe can easily be applied to existing instruments and data. So long as the original scan files have been retained, data collected from previous deployments and field campaigns can be reprocessed using this technique to reveal latent information that has not yet been seen. Thus, additional effective range from existing CDL infrastructure can be realized with no additional capital expense.

*Data availability:* DL and Sonde data from ARM SGP Central Facility is available via ARM Data Center (https://adc.arm.gov/discovery/#/)

*Author contributions:* SB, DDT, and WAB developed the retrieval code. SB, TJW and DDT analyzed the data and wrote the manuscript.

*Competing interests:* The authors declare that they have no conflict of interest.

*Acknowledgements:* This research was supported in part by NOAA cooperative agreements NA17OAR4320101 and NA22OAR4320151, the U.S. Department of Energy's Atmospheric System Research, an Office of Science Biological and Environmental Research program, under Grant No. DE-SC0020114, and the NOAA Atmospheric Science for Renewable Energy Program.

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

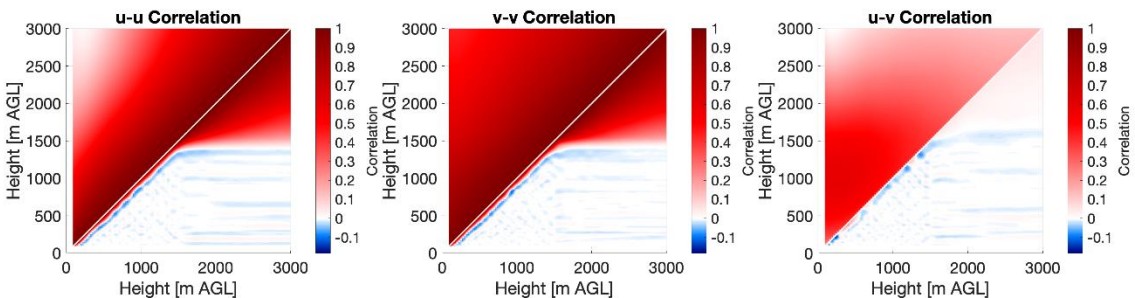

**Figure 1: Correlation matrices for *u-u* (left), *v-v* (center), and *u-v* (right) for the month of July. The upper left half of each panel shows the correlation of the *a priori*, while the lower right half shows the correlation of the posterior for a clear-sky retrieval at 0235 UTC on 16 July 2019.**

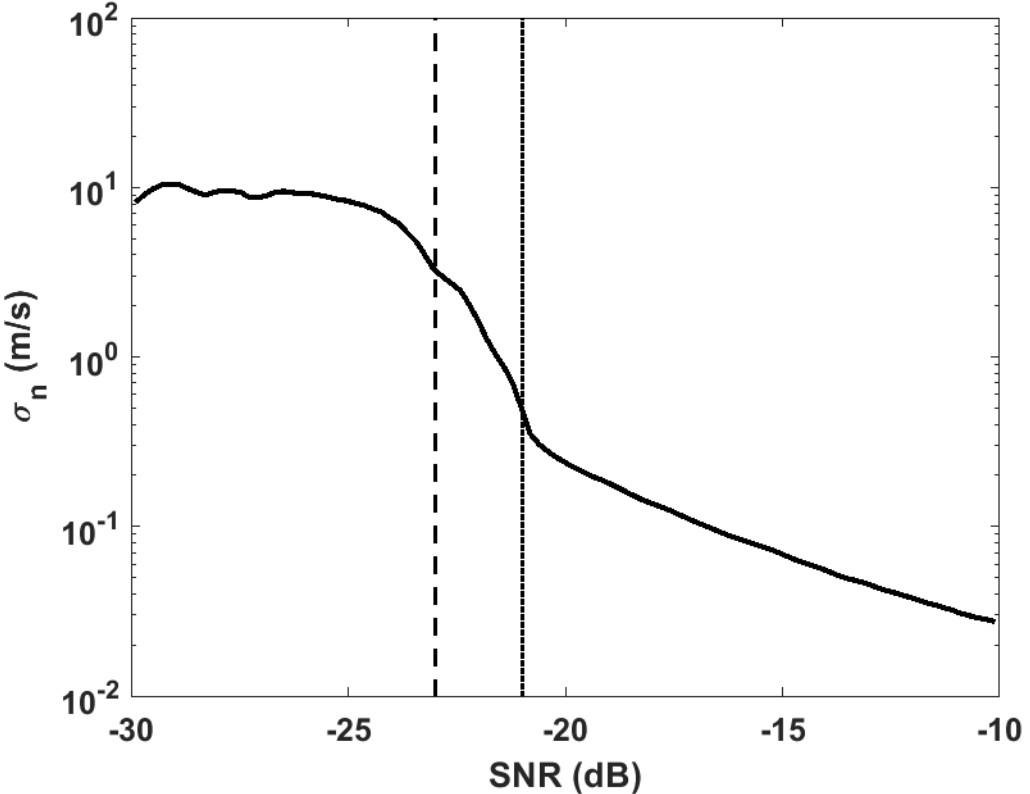

**Figure 2: Measurement precision as a function of SNR for a day showing different SNR thresholds applied for VADtrad (dotted line) and VADoe retrievals (dashed line).**

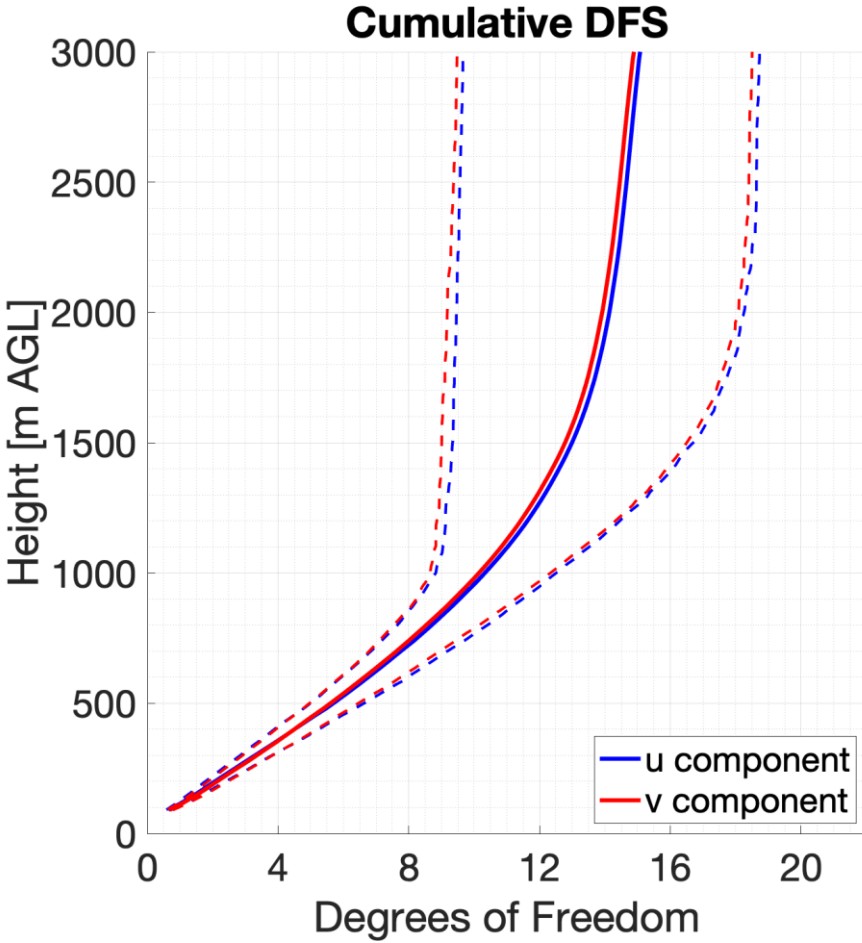

**Figure 3.** **Vertical profile of the mean (solid) and 25th and 75th percentile (dashed) cumulative degrees of freedom of the signal calculated from the OE wind retrieval for both the *u* component (blue) and *v* component (red).**


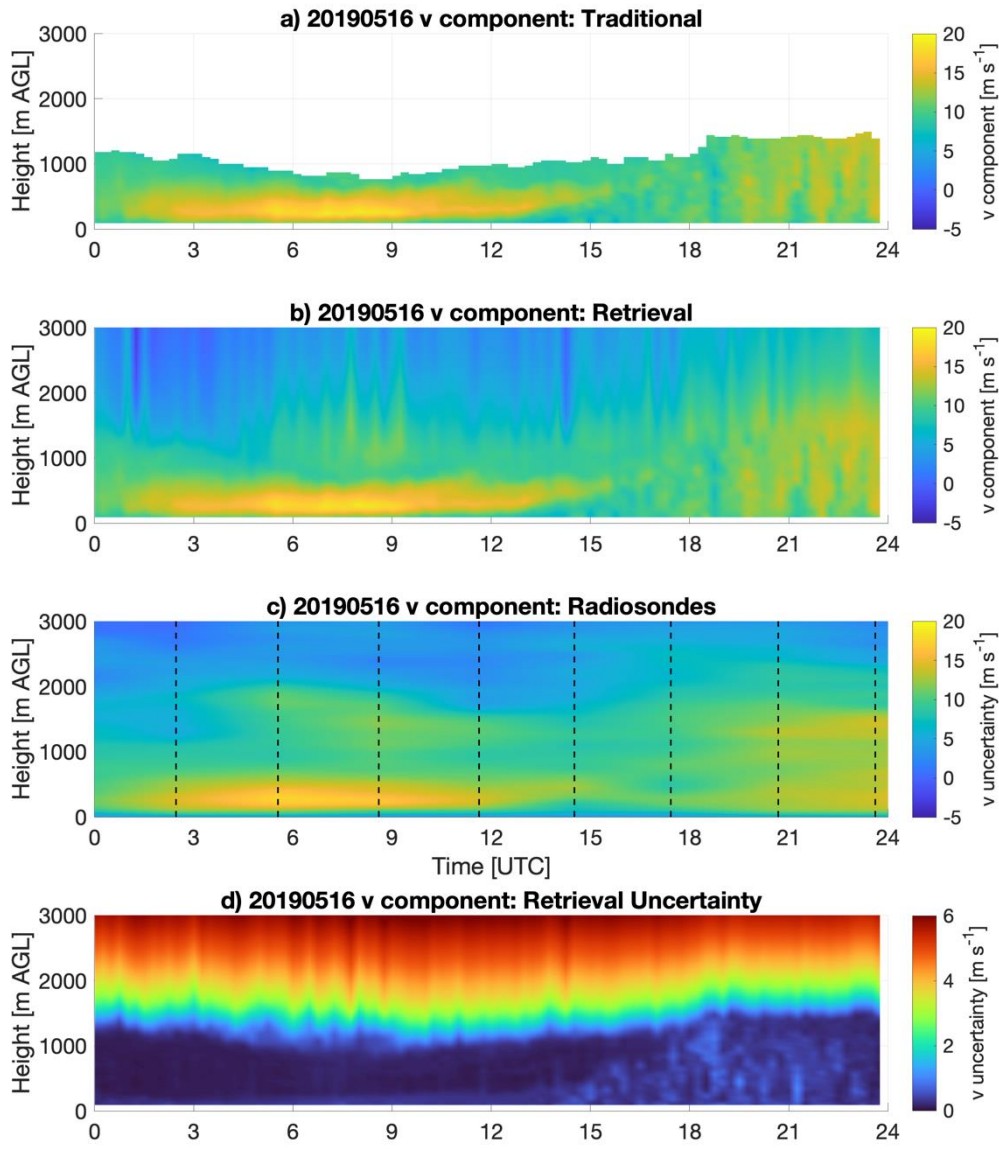

**Figure 4: Time-height cross sections of the *v* component of the wind on 16 May 2019 as observed by VADtrad (a), VADoe (b), and radiosonde (c). The uncertainty in the VADoe retrieval is shown in panel d in a different color scale to enhance detail. Radiosondes were launched every 3 h at the times indicated by dashed lines in the third panel. Radiosondes data are interpolated in time for illustration purposes. Time is in UTC; local time is UTC - 5.**

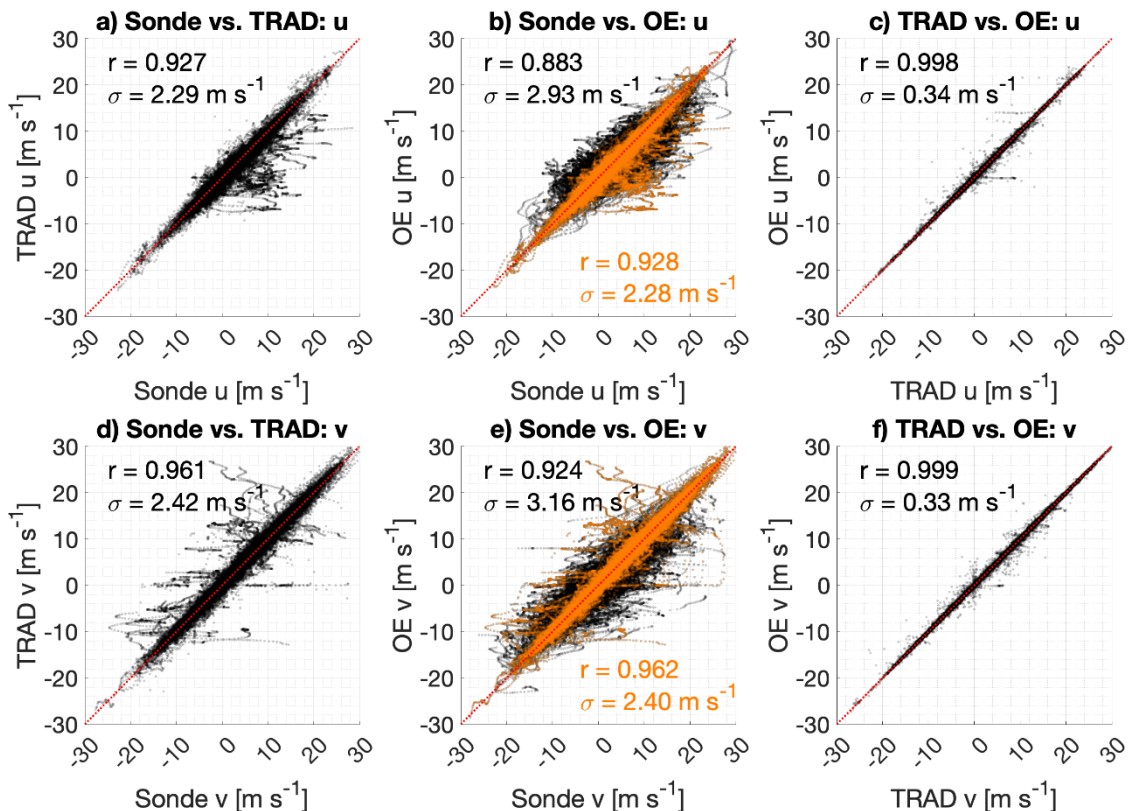

**Figure 5: Scatter plots of the *u* component (top row) and *v* component (bottom row) of wind for radiosonde vs. VADtrad (left column, N= 59,403), radiosonde vs. VADoe (center column, N=139,582), and VADtrad vs. VADoe (right column). The dotted red line represents the 1:1 line. Points in orange indicate the subset of VADoe observations for which a valid VADtrad observation also exists.**


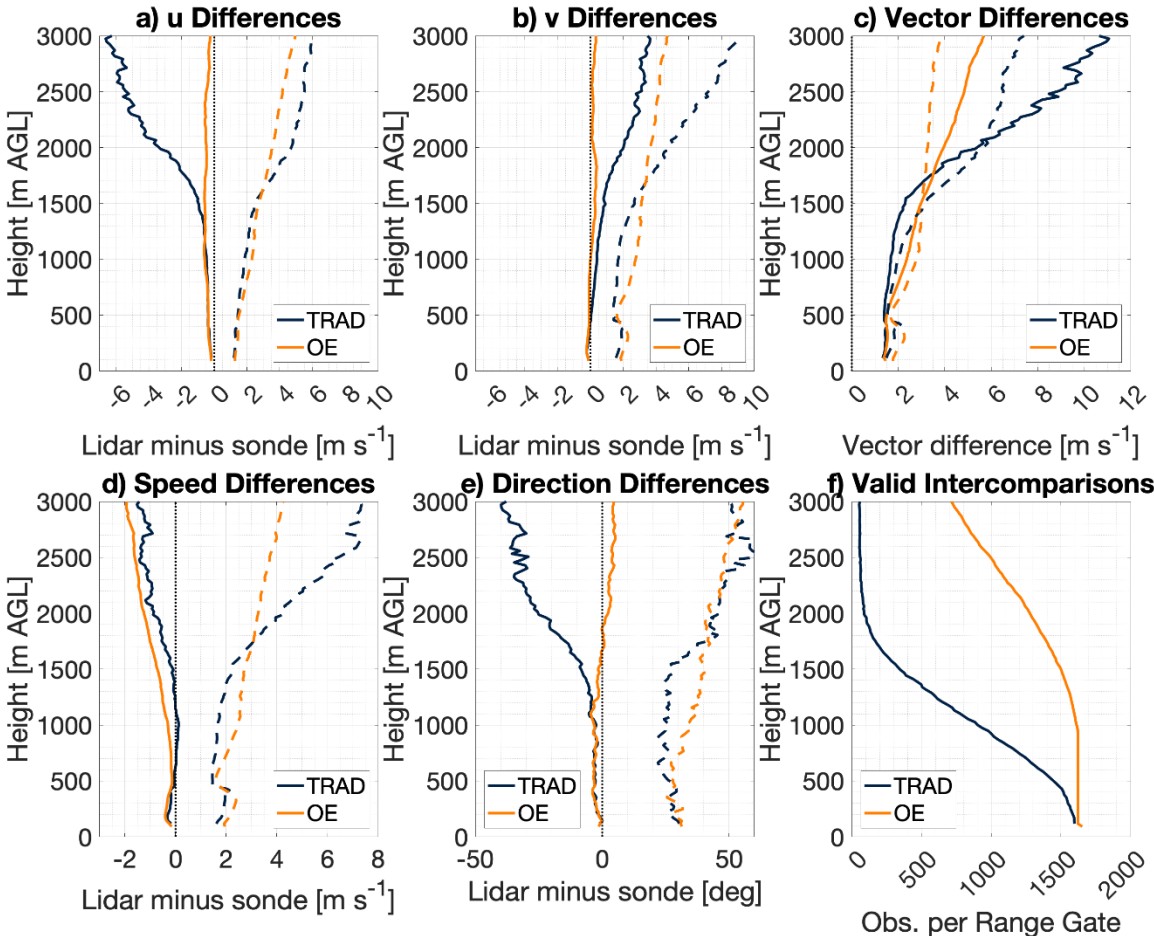

**Figure 6: Vertical profiles of the bias (solid line) and 1-sigma uncertainty (dashed line) for VADoe (orange) and VADtrad (dark blue) for a)** *u* **winds in m s⁻¹ , b)** *v* **winds in m s⁻¹, c) wind vector differences in m s⁻¹, d) wind speed in m s⁻¹, and e) wind direction in deg. Panel f) shows the number of valid sonde versus lidar intercomparisons as a function of height for both algorithms.**

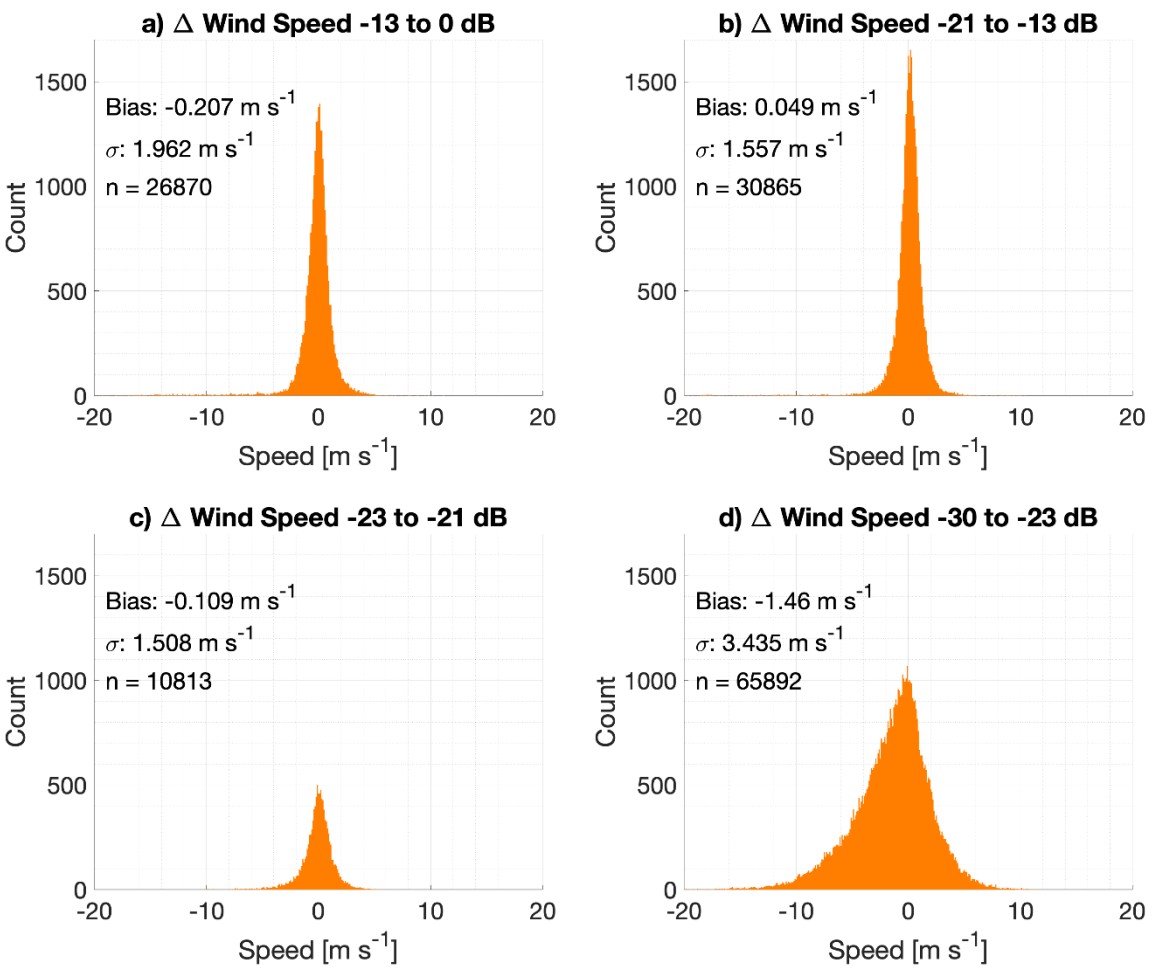

**Figure 7: Histograms of the VADoe retrieval minus radiosonde differences in wind speed (in m s[-1]) for four different bands of lidar signal to noise ratio.**

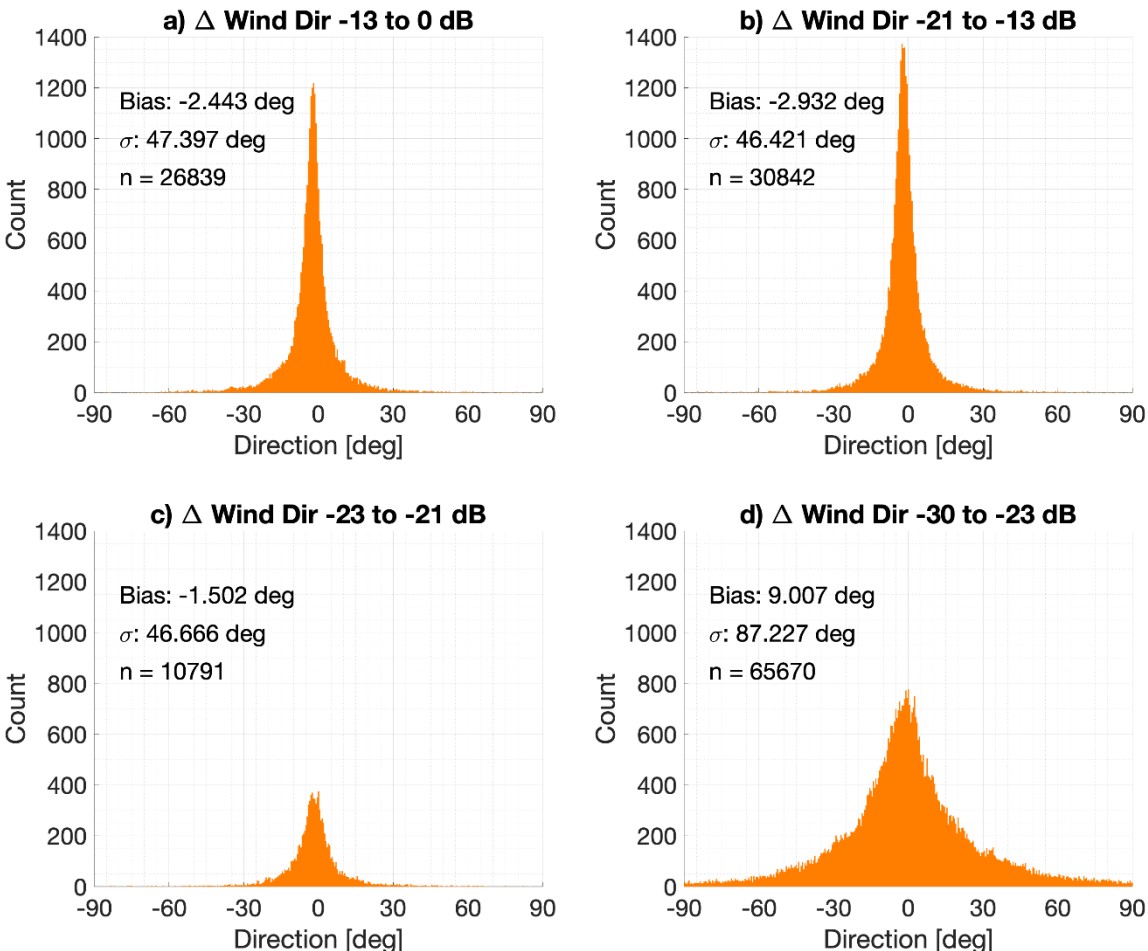


**Figure 8:** As in Fig. 7, but for wind direction (in degrees).

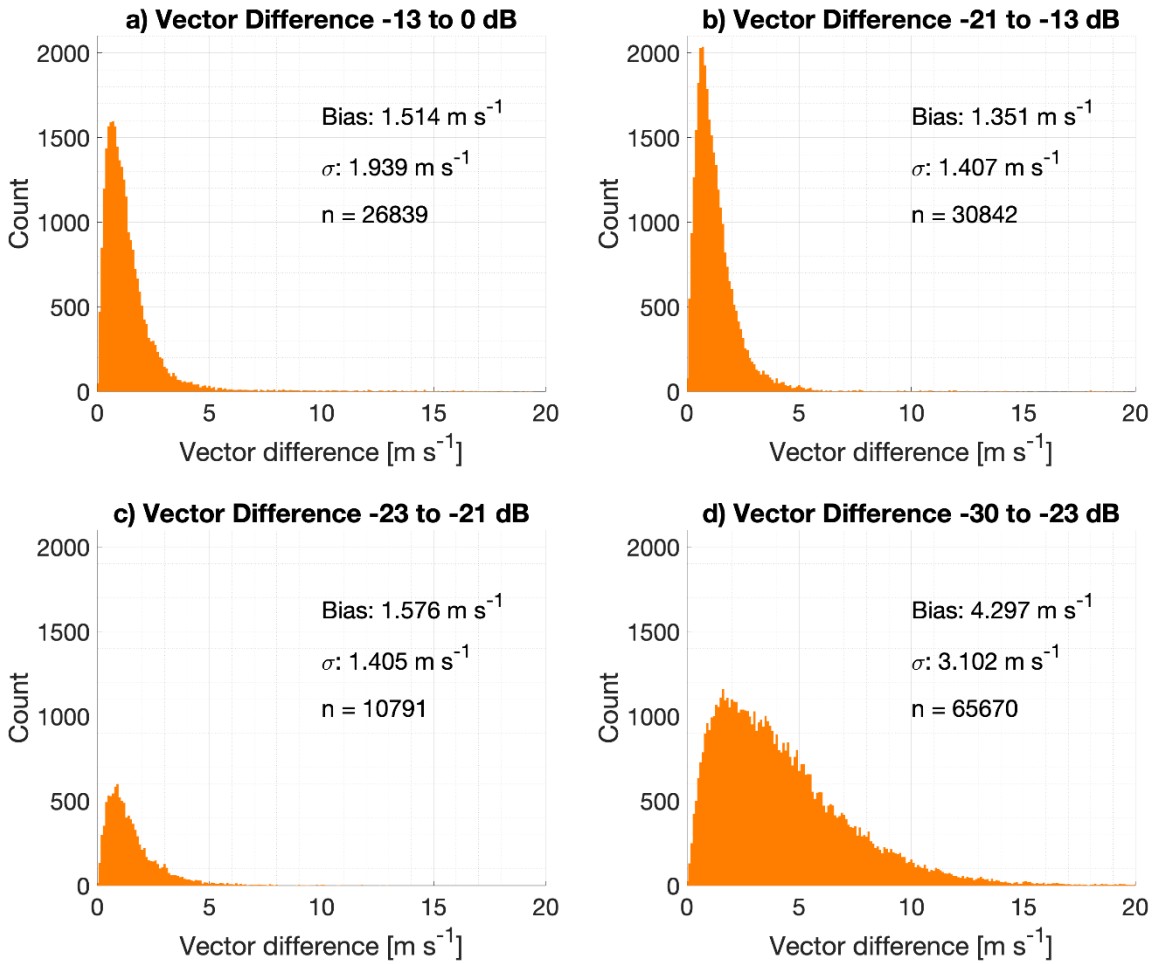

**Figure 9:** **As in Fig. 7, but for vector difference (in m s⁻¹).**