# Peer review of "Using Optimal Estimation to Retrieve Winds from VAD Scans by a Doppler Lidar"

_Atmospheric Measurement Techniques, 2022_

## Referee Comment (RC2)

Baidar et al

Using Optimal Estimation to retrieve winds……

20 March 2023

Major Revision and then acceptance

General

This work uses lidar observations based on VAD and VADoe methods and compare the results with radiosonde observations. VAD technique is well known for many years. Optimal estimation (OE) is a new technique for wind analysis and claimed to be better than VAD alone. VADoe advantage is to have full covariance analysis at each level improve the wind profiles. It is claim that OE technique provide wind data at the levels where classical VAD not but uncertainty is very high. It is suggested that new method is better than old one and provided data without having any instrument hardware changes.

Major issues

Overall, what is claimed to be can be useful but with large uncertainty; this means the results can have very large uncertainty at the higher levels. If you compare the figs 5 and 6, as well as 4, you can see what is going at higher levels.

Ln 29; Gultepe et al 2018 A review on aviation meteorology…. PAAG) can be provided here.

Ln37/60; seems these parags should be given under the method section. Intro is very short if these parags are taken to another section. Intro should be developed into a better summary of earlier obs/issues.

Fig. 4;  how can we say the retrieval (middle) provided better results compared to radiosonde; I see the strikes in the retrievals. VAD shows nothing. How can say that retrievals are better?

Fig 5; Compare VAD versus OE; OE shows much larger scatter of the data points compared to VAD ones. Then how can we say OE led to better results compared to VAD?

Fig. 6; OE shows better results (a) but with large uncertainties (Fig. 5). Sd for both methods are bad at higher levels any way. Again issues exist at higher levels and OE technique provides bad results (higher error) but VAD provides no results. Based on this  can we use OE results accurately? Probably not. Say if wind speed 2 m s-1 and error is 4 m s-1, then why we have to use this data?

Fig 5 and 6; how many data points are used in the analysis and how data is averaged? Please provide some info in the captions of Fig5 and 6.

Please do a Discussion section before the Conclusions. Then discuss the issues mentioned above.

Conclusions:

- please provide a description of VAD technique, and what assumptions used?
- What equation is used in VAD? What was the vertical air motion at the surface? Zero? Right? Was it correct?
- …..with correlations of 0.998 and 0.999 between the VADtrad and VADoe for u and v, respectively?Is this correct? If they are 100% correlated, it means no difference between 2 methods, then why we need OE technique? This cant be correct because OE provided large uncertainty compared to VAD when compared to radiosonde. Why is that?
- Results represent what? 1 day or 6 months? Also why not provide a few extreme cases?
- Ln353/354 is this correct? Please provide the conclusions with bullets..

---

## Author Comment (AC1)

We thank the reviewer for the helpful comments and suggestions. In the our response to the reviewer comments, the Reviewer Comment is first reproduced in black, followed by our response in blue, and changes to the manuscript in *orange*.

This manuscript presents a novel method to apply optimal estimation techniques to retrieve the wind profile continuously up to 3 km using Doppler lidar measurements.  This overcomes a main limitation in current Doppler lidar wind measurements that are typically limited by the presence of aerosols and clouds in the lower atmosphere.  By providing continuous measurements up to 3 km with this method, error covariances are also created facilitating assimilation of the wind profiles into NWP models.  The benefits of this new technique over current approaches will be of high interest to readers of AMT, especially those in the remote sensing and NWP communities. However, there are a number of issues that need to be addressed, detailed below, prior to this being fully acceptable to AMT.  Most notably, additional analysis (which should be quickly and easily performed) will be necessary to support important claims that VADoe wind retrieval errors meet WMO standards for use in NWP.

Specific Comments

1. Line 23: In addition to providing wind speed uncertainty in the abstract, it would be best to provide the vector RMSE to account for wind direction uncertainty for applications where the wind direction is important (e.g., storm mode forecasting, aviation).

   We have added vector RMSE to the text as suggested.

   We noticed that not all the analysis presented in the paper included the 5 m/s VADoe error threshold filtering criteria described in the paper. We have corrected this in the revised manuscript. Any changes you see with regards to figures and statistics are related to this update

   *… OE wind speed and wind vector have uncertainties of 3.44 and 4.33 m/s respectively.*

2. Line 44: Suggest changing 'stares' to 'measures' or 'points', as stares imply the lidar may be pointed in that given position for a prolonged period of time which may not always be true.

   We have changed 'stares' to 'measures' as suggested.

3. Line 56: In addition to the proposed VADoe method here, there are other novel methods (e.g., Stephan et al 2019) that can be used to extend the range of lidar wind measurements that should also be referenced and discussed at least in the introduction, perhaps elsewhere. They provide enhanced range compared to VAD and likely lower error statistics than VADoe, but will not provide continuous measurements to 3 km.

We have added following text to the introduction.

*Various techniques have been developed to extend the range of wind profiles from scanning CDL, including accumulation of signal power spectra estimates for direct estimation of the wind vector without estimating radial wind velocities for individual azimuth angles (Smalikho, 2002; Stephan et al., 2019). Although these advanced techniques are able to extract information from noisier Doppler spectra, they are still limited by the availability of the scatterers and hence, do not provide consistent vertical coverage.*

4. Section 2.2: While this section is a nice detailed explanation of the VADoe technique, there are some questions that remain. Specifically, how is the VADoe technique applied when there are few or no valid measurements made from the lidar, such as when low-clouds and fog completely attenuate the signal within the lowest tens of meters? Is a retrieval still made? If so, should one be made? If not, how much 'valid' data (measurements above -23 dB) is needed to make a retrieval?

VADoe retrievals are still made even when there are very few or no valid measurements are available for the Doppler Lidar. In such cases, the retrieved profile will follow the a priori profile, and it can be identified using retrieval errors. The figure below (Fig. R1) shows example cases when there was a period with no valid data (red). Profiles are 3 hours apart as radiosondes profiles were only available every 3 hours.

[Figure]

Figure R1: Profiles of (left) snr, (middle) u and v, and (right) u and v error for three different time periods. The red profile is an example of a case when no valid CDL data were available. The green profiles in the middle panel shows the a priori profiles.

We have added following text to manuscript for clarification.

*In cases when there are very few or no valid CDL measurements (e.g. a very low aerosol loading, foggy or rainy day), the retrieved profiles are only constrained by the a priori and hence, follows the a priori profile. Such profiles can easily by identified using the averaging kernel (A) matrix, DOF or retrieval error.*

5. Line 207: Were any other simple quality control measures applied to ensure there were not significant changes in the wind between the radiosonde launch and the lidar measurements (e.g., front passages, convective outflows, etc)? While I'm sure out of the large dataset, there's only small fraction of instances when that may have happened given the lidar profiles are generally <8 min from the radiosonde launch, but these cases may have an outsized effect on later statistics presented. A simple filter looking for large differences in wind speed and direction throughout the sonde and lidar wind profiles could detect these cases.

No, we did not apply any filtering conditions to exclude extreme cases. We did a gross error check by filtering any VAD wind observations with an absolute value greater than 50 m/s assuming that they were unphysical. This information is already included in the manuscript (line 256). We have done the analysis by applying a 3-sigma filter to exclude the extreme outliers instead of the 5 m/s error threshold, and the results looked very similar. Figure R2 is same as Figure 5 in the manuscript but with 3-sigma filter. The correlation coefficient for sonde vs VADtrad, and sonde vs VADoe improved a little for cases where valid VADtrad and VADoe observations were available. There was almost no change for the overall sonde vs VADoe comparison. Thus, we decided to not implement additional filtering conditions to exclude extreme cases.

[Figure]

Figure R2: Same as Figure 5 in the manuscript with 3-sigma filtering applied.

6. Figure 3: Personally, I find the representation of the 1-sigma confusing, particularly when trying to compare what is shown here with what is discussed in the next (DOF ranges from 4.9 to 25.3). It would be better to show the 1-sigma as error bars around the solid line showing the mean. An alternative option would be to add multiple dashed lines each representing the mean +/- 1 sigma.

We have replaced Figure 3 with the figure shown below, which shows the range of DOF as each altitude. We also replaced the +/- 1 sigma with 25th and 75th percentile since standard deviation show decrease in Cumulative DFS at higher altitude due to having fewer points.

[Figure]

*Figure 3: Vertical profile of mean (solid) and 25th and 75th percentile (dashed) cumulative degrees of freedom of the signal calculated from the OE wind retrieval for both the u component (blue) and v component (red).*

7. Figure 4: It would be helpful to add an additional panel to show the errors associated with the OE retrieval of v. This would help the reader understand the accuracy of the wind estimate, particularly above the PBL where I assume the magnitude of the vertical striping is within the larger uncertainty of the retrieval at those higher altitudes.

We have added a fourth panel showing errors for OE retrievals of v. We used a different colormap for the error figure to avoid confusion with the other time-height cross-sections.

[Figure]

*Figure 4: Time-height cross sections of the v component of the wind on 16 May 2019 as observed by VADtrad (a), VADoe (b), and radiosonde (c). The uncertiainty in the VADoe retrieval is shown in panel d in a different color scale to enhance detail. Radiosondes were launched every 3 h at the times indicated by dashed lines in the third panel. Radiosondes data are interpolated in time for illustration purposes. Time is in UTC; local time is UTC - 5.*

For most part the profile to profile difference in v, |δv| (and u, not shown) is smaller than the OE retrieval error (see Figure R3). Note that the larger |δv| inside the PBL, compared to retrieval error, is due to natural temporal variability and turbulence.

[Figure]

Figure R3: Profile of mean (solid) and 1-sigma (shaded region) absolute profile-to-profile difference of v component (red) and v component retrieval error (blue).

*Results at higher altitudes are more influenced by nearest good measurements compared to further away. The profile-to-profile difference at higher altitudes are within VADoe retrieval error for most cases as shown in Fig. 4d.*

8. Line 245: Additional clarification is needed here, likely requiring rewording. Are the radiosonde observations (every 3/6 hr) interpolated to a 15-min resolution for comparison of wind? If that's the case, this is not a good approach as there can be significant errors in interpolating over 3/6 hr gaps, and the comparison should be done by bilinearly interpolating the lidar observations around a radiosonde launch to the launch time (at the VADtrad measurement heights).

Radiosondes observations were available every 3 hours. Each radiosonde was temporally matched to the Doppler lidar profile that was taken nearest in time to the radiosonde launch time. If the closest valid Doppler lidar profile was more than 30 min from the radiosonde launch time, it was excluded from this analysis. This information is included in the paper in line 208-210 (now lines 212-214).

Radiosondes data were averaged to the same vertical grid as VADtrad and not time grid. We have changed "grid" to "vertical grid" for additional clarity. Radiosonde data were interpolated in time in Figure 4c for illustration purposes only.

*Note that radiosonde profiles shown in Fig. 4 are interpolated in time for illustration purposes.*

*To facilitate intercomparisons between the radiosondes and both VADtrad and VADoe, the same vertical grid from the traditional VAD technique was used for the OE output, and the radiosonde observations were interpolated averaged to that grid.*

Line 248: While the OE retrieved profiles are inherently smooth, it's not fair to smooth the radiosonde profiles for the comparison with the OE retrievals but not for the VADtrad measurements. By smoothing the radiosonde profiles for the OE-retrievals, the error statistics are likely going to biased low (showing better performance than the OE retrievals actually perform when compared with observations, given inherent limitations of the OE method), misleading readers. The radiosondes should not be smoothed for either comparison.

We agree with the reviewer that radiosonde profiles should not be smoothed for comparison with OE retrievals. We actually did not use Averaging Kernel smoothed radiosonde profiles for comparison with OE retrievals as mentioned in the text. We have removed that text from the manuscript. Radiosonde data are available in higher vertical resolution than the Doppler lidar range gates. So, all of the sonde values within a given range gate are averaged together to obtain a representative sonde value for that range gate. Please refer to the reply to the previous comment for changes to the manuscript.

9.  Line 251: Wind precision estimates are available for the VAD profiles (as stated at line 77). Why are they not used here, with a similar criterion of rejecting data wherein the uncertainty exceeded 5 m/s?

    VADtrad from ARM database does not provide estimates for snr <-21 dB, and hence the errors are much smaller. There are actually no data points with error > 2 m/s, and hence the same filtering criteria was not used. We have added this information to the manuscript for clarity.

    *Note that due to the stringent SNR threshold (<-21 dB) applied to the VADtrad data from the ARM database, there were no VADtrad observations with uncertainty greater than 5 m s$^{-1}$.*

10. Lines 328: The bias of the wind speed measurements of VADoe in the low SNR band is considerably worse than the referenced TAMDAR paper (bias of 0.90 m/s, vs -2.52 m/s for VADoe). I do not agree they are comparable as stated.\

We respectfully disagree that the reviewer. Since larger bias is easily correctable and is not as significant of an issue as a larger random error (as quantified by the standard deviation), our statement only compared uncertainties for the TAMDAR and VADoe in the lowest SNR bin, which are indeed comparable. In fact, VADoe uncertainty is better compared to TAMDAR without 3 σ check (4.49 m/s for VADoe vs 6.44 m/s for TAMDAR). We have updated our analysis by applying 5 m/s VADoe retrieval error filter to the consistent with the rest of the manuscript. The new bias and uncertainty for the lowest SNR band is -1.46 m/s and 3.44 m/s respectively. This is comparable to TAMDAR uncertainty of 3.37 m/s after 3 σ check. Hence, we have left the statement as it is.

11. Line 330: Insufficient results are presented to support the claim here that the VADoe measurements at the low SNR band meet the WMO threshold requirement for horizontal wind in the free troposphere for global and high-resolution NWP. The WMO requirements referenced are given as the vector error in m/s. The authors do not present the vector error, but instead only present results for the wind speed and wind direction separately. The vector error will be a combination of these, and will be considerably worse than the presented wind speed error alone. In order to ensure that the VADoe wind retrievals at the lower SNR are acceptable for assimilation into NWP following WMO standards, the authors must also present result showing the performance of VADoe vector RMSE. This will require the additional analysis and another set of figures, similar to Figures 7/8, with a supporting discussion.

We have add new wind vector difference analysis as suggested (Figure 9). The wind vector RMSD for the lowest SNR band is 4.3 m/s, which is within the 5 m/s WMO threshold requirement. We have also modified the statement to highlight that VADoe error could be used as filtering criteria to select that meet different application requirements.

[Figure]

Figure 9: As in Fig. 7, bur for vector differences (in m/s).

*The wind vector RMSD of less than 5 m/s, which is the error threshold used in the analyses, for this SNR group further supports that the VADoe retrieval errors are representative, and can be used to select data to meet different application requirements. For example, VADoe data filtered for greater than 5 m/s error would meet the WMO threshold requirement for horizontal wind measurements in the free troposphere for Global and high resolution NWP (WMO, 2022).*

12. Line 363-365: Similar to the two above comments, this statement must be removed or further supported with additional data analysis. The VADoe estimates at the additional effective range appear to be significantly worse than TAMDAR and may not meet WMO threshold requirements for NWP.

We have removed this statement from the conclusion as it could be wrongly interpreted as all VADoe data would meet these requirements. However, we have added analysis/figures to support this statement earlier in the paper. Please see reply to the previous two comments for details.

Reference

Stephan, A., Wildmann, N. and Smalikho, I.N., 2019. Effectiveness of the MFAS method for determining the wind velocity vector from windcube 200s lidar measurements. *Atmospheric and Oceanic Optics*, *32*, pp.555-563.

This reference has been added to the paper along with Smalikho (2003).

*Smalikho, I.: Techniques of wind vector estimation from data measured with a scanning coherent Doppler Lidar, J. Atmos. Ocean. Technol., 20(2), doi:10.1175/1520-0426(2003)020<0276:TOWVEF>2.0.CO;2, 2003.*

---

## Author Comment (AC2)

We thank the reviewer for the helpful comments and suggestions. In the our response to the reviewer comments, the Reviewer Comment is first reproduced in black, followed by our response in blue, and changes to the manuscript in *orange*.

General

This work uses lidar observations based on VAD and VADoe methods and compare the results with radiosonde observations. VAD technique is well known for many years. Optimal estimation (OE) is a new technique for wind analysis and claimed to be better than VAD alone. VADoe advantage is to have full covariance analysis at each level improve the wind profiles. It is claim that OE technique provide wind data at the levels where classical VAD not but uncertainty is very high. It is suggested that new method is better than old one and provided data without having any instrument hardware changes.

Major issues

Overall, what is claimed to be can be useful but with large uncertainty; this means the results can have very large uncertainty at the higher levels. If you compare the figs 5 and 6, as well as 4, you can see what is going at higher levels.

We agree with the reviewer's assessment that the VADoe results have large uncertainty at higher levels where traditional VAD do not provide any results. However, we disagree with the reviewer's assessment that higher uncertainty results are inherently not useful. VADoe provides the mostly likely wind conditions at those levels based on the (noisy) observations at that height winds at lower levels, where CDL makes accurate measurements, and climatology. VADoe retrieval provides uncertainty estimates for each output, and users can decide the uncertainty threshold based on their application requirement. This is an improvement compared to traditional VAD which do not provide any output for SNR below a certain threshold. Figure 7c and 8c clearly shows the benefit of VADoe as some very accurate measurements are discarded by VADtrad due to this hard SNR cutoff. In addition, VADoe more easily facilitates assimilation into NWP as it provides uncertainties, and averaging kernels needed for data assimilation.

Ln 29; Gultepe et al 2018 A review on aviation meteorology…. PAAG) can be provided here.

This reference has been added to the paper.

Ln37/60; seems these parags should be given under the method section. Intro is very short if these parags are taken to another section. Intro should be developed into a better summary of earlier obs/issues.

We respectfully disagree with the reviewer on moving this section to the method section. This section describes the CDL observation and associated issues, why the measurements are limited to lowest 1-2 km, and a need for a new method that could provide consistent vertical coverage. Thus, this section is better suited in the introduction.

Fig. 4; how can we say the retrieval (middle) provided better results compared to radiosonde; I see the strikes in the retrievals. VAD shows nothing. How can say that retrievals are better?

We do not claim that the VADoe retrieval provides better results compared to radiosondes. We use radiosondes as a reference to determine how well the VADoe performs at different levels. We also do not claim that VADoe provides better results than the traditional VAD method. What we do claim is that VADoe provides equivalent results as VADtrad where VADtrad provides an output. Where the VADTrad does not provide results, VADoe provides the statistically most likely results based on the (noisy) observations at that height, the higher-signal measurements at lower levels, and climatology. Thus, there are only benefits to using VADoe.

Fig 5; Compare VAD versus OE; OE shows much larger scatter of the data points compared to VAD ones. Then how can we say OE led to better results compared to VAD?

We respectfully disagree with the reviewer's assessment that VADoe shows much larger scatter compared to VADtrad. Figure 5c and 5f compares VADtrad and VADoe for u and v vectors respectively and the scatter in data is comparable. The larger standard deviation seen in Figure 6a-e is due to the larger number of data points included in the comparison. Please see Figure 6f for number of data points included in the comparison at each level.

We have added standard deviation information to Figure 5 depicting the scatter in data points.

Fig. 6; OE shows better results (a) but with large uncertainties (Fig. 5). Sd for both methods are bad at higher levels any way. Again issues exist at higher levels and OE technique provides bad results (higher error) but VAD provides no results. Based on this can we use OE results accurately? Probably not. Say if wind speed 2 m s-1 and error is 4 m s-1, then why we have to use this data?

We would characterize OE results at higher levels as results with higher uncertainty rather than "bad results". The beauty of the OE retrieval is that it provides propagated error for each result based on measurements and a priori error covariances. It also provides the Averaging Kernel matrix which provides information about vertical resolution as a function of height as well as the maximum height to which the retrieval is mostly independent of the a priori profile. So, users can decide what they want to consider as good or bad data depending upon application. This is currently not an option with traditional VAD retrieval.

Fig 5 and 6; how many data points are used in the analysis and how data is averaged? Please provide some info in the captions of Fig5 and 6.

We have added number of data points used in Fig 5 in the figure caption (N = 59,403 for trad vs sonde, and N = 139,582 for oe vs sonde). Radiosonde data were vertically averaged to lidar vertical grid and temporally matched to lidar (within 30 mins) for this analysis. This information is already included in the paper (lines 212-214).

Figure 6f shows number of data points at each level included in the intercomparison.

*Figure 5: Scatter plots of the u component (top row) and v component (bottom row) of wind for radiosonde vs. VADtrad (left column, N= 59,403), radiosonde vs. VADoe (center column, N=139,582), and VADtrad vs. VADoe (right column).*

Please do a Discussion section before the Conclusions. Then discuss the issues mentioned above.

We have added a discussion section as suggested.

*The comparisons in the previous sections show that VADoe provides identical results as VADtrad where VADtrad results are valid. At these levels, where most if not all of the information are coming from measurements, VADoe is mathematically equivalent to VADtrad. At lower SNR levels (or higher altitudes), where VADtrad results are not available, VADoe results compare favorably with radiosonde measurements. VADoe at those levels are statistically most likely output based on the (noisy) observations at those levels, higher quality (precision) measurements at lower levels and climatology. The VADoe retrieval provides well characterized uncertainty for each profile, and the corresponding averaging kernels allow the determination of both the vertical resolutions as a function of height and the maximum height to which the retrieval is mostly independent of the a priori profile. Thus, the retrieval errors and averaging kernels could be used to determine data that are suitable for a given application.*

*One of the biggest challenges of setting up the VADoe retrieval is appropriately scaling the CDL radial velocity measurement error at low SNRs to provide stable retrievals. The CDL radial velocity measurement error is limited by the measurement bandwidth. For example the measurement bandwidth for the ARM SGP CDL used here is +/- 19 m/s. This maximum measurement error is smaller than the a priori error (standard deviation). This becomes even smaller when you consider multiple radial velocities from different azimuths that are included in the retrieval. If the measurement errors are not inflated appropriately, measurements will always be weighted heavily compared to a priori, and results in unstable retrievals. Thus, measurement error at low SNR levels needs to be appropriately scaled accounting for number of azimuths and elevation angles included in the retrieval and magnitude of the a priori error.*

*Successful implementation of VADoe retrieval requires knowledge of the a priori mean profile and covariance. We used radiosonde measurements to create monthly mean profile and covariance. However, radiosonde measurement sites are limited which limits the applicability of the VADoe retrieval presented here to locations close of radiosonde sites. Future work should include testing using a priori from other sources such as AMDAR and NWPs. This would make VADoe retrieval more widely applicable, and also use of higher time resolution a priori.*

Conclusions:
• please provide a description of VAD technique, and what assumptions used?

Description of the VAD technique and assumptions used are provided in the section 2.1 Traditional VAD Method (VADtrad). We have also added equation used in VAD.

• What equation is used in VAD? What was the vertical air motion at the surface? Zero? Right? Was it correct?

We have added the equation used in VAD. Please see new Eq. (1).

Vertical air motion at the surface is not measured by ground based CDL. Typically CDLs have a blind zone near the instrument that is twice the laser pulse length.

• …..with correlations of 0.998 and 0.999 between the VADtrad and VADoe for u and v, respectively?Is this correct? If they are 100% correlated, it means no difference between 2 methods, then why we need OE technique? This cant be correct because OE provided large uncertainty compared to VAD when compared to radiosonde. Why is that?

Yes, the correlations between VADtrad and VADoe for u and v are 0.998 and 0.999 respectively (see figure 5c and 5f), and there is no difference between the two methods where VADtrad provides the results. What this excellent agreement shows is that there is no disadvantage in using the VADoe compared to VADtrad. The key point of this work is that VADtrad does not provide results at higher levels where the measurement SNR is below a certain threshold. The VADoe provides additional information at levels where VADtrad does not provide any results without any loss of information where VADtrad is valid.

Larger standard deviation for VADoe relative to VADtrad when compared to radiosonde that is seen in Figure 6 is due to the larger number of data points included in the comparison for VADoe. Figure 6f shows number of data points included in the comparison at each level. For example, there are twice as many data points included in the VADoe comparison compared to VADtrad.

• Results represent what? 1 day or 6 months? Also why not provide a few extreme cases?

The results presented in the paper are for the entire 2019 calendar year. This information is already included in the paper. We have also added this information to the abstract and conclusion.

It would be very difficult to assess the performance of VADoe retrieval under extreme cases because the reference sonde profiles themselves might not be representative. Thus, we did not highlight extreme cases. But we also did not exclude any extreme cases. One could also consider all data at low SNR levels are extreme cases.

*This method was tested using a yearlong CDL measurements in ARM SGP Central Facility in 2019*

• Ln353/354 is this correct? Please provide the conclusions with bullets..

Yes, it is correct. Figure 4-8 shows results at higher altitudes where SNR is too small and radial velocities are not reliable. As mentioned previously, VADoe results at those levels are statistically most likely results based on the noisy observations at those levels, higher quality measurements at lower levels and climatology. Demonstrating the application of the VADoe product is beyond the scope of this paper.